# Motor–Cognitive Interventions May Effectively Improve Cognitive Function in Older Adults with Mild Cognitive Impairment: A Randomized Controlled Trial

**DOI:** 10.3390/bs13090737

**Published:** 2023-09-04

**Authors:** Mingda Tao, Huajun Liu, Jinxuan Cheng, Caiyun Yu, Lili Zhao

**Affiliations:** Normal College, Qingdao University, Qingdao 266071, China; mingdatao@qdu.edu.cn (M.T.); lhjrose@163.com (H.L.); yucaiyun98@163.com (C.Y.); lilizhao@qdu.edu.cn (L.Z.)

**Keywords:** older adults, mild cognitive impairment, cognitive ability, motor–cognitive intervention, prevention

## Abstract

Mild cognitive impairment (MCI) is a syndrome that occurs in the preclinical stage of Alzheimer’s disease. Early intervention can be effective in preventing Alzheimer’s disease, but further research is needed on intervention methods. To identify interventions that are more suitable for Chinese characteristics and to investigate the effects of motor–cognitive intervention on the cognitive functions of older adults with MCI, we screened 103 community-dwelling older adults with MCI aged 65 years and older in Qingdao, Shandong, China; divided them into an intervention group and a control group; and administered a motor–cognitive intervention to the intervention group for 12 weeks. The study used the Mini-Mental State Examination (MMSE) and the Montreal Cognitive Assessment (MoCA) to assess the initial cognitive level of the MCI participants and detect the effects of the intervention. We found that the cognitive abilities of the intervention group were significantly improved at the end of the intervention, as well as at the end of the follow-up, compared with the control group. The results of the current study suggest that the motor–cognitive intervention we used may improve the cognition of older people with MCI in the Chinese community.

## 1. Introduction

The trend of population aging is growing. The results of relevant studies show that, by 2050, the number of older adults in the world will reach three times the current population [1]. The health status and behavior of older people are of great interest to researchers, and cognitive impairment is a common problem among older people. Mild cognitive impairment (MCI) is a syndrome that occurs in the preclinical stage of Alzheimer’s disease (AD) and is a condition in which individuals exhibit cognitive impairment with minimal impairment in the instrumental activities of daily living [2]. The variability in the prevalence of mild cognitive impairment is already more than five times greater in Asian countries. The prevalence of MCI is increasing in both developed and developing countries as the aged population increases. The World Health Organization has made prevention and control related to the development of Alzheimer’s disease prevention strategies an international priority [3]. MCI is the intermediate stage between cognitive decline caused by normal aging and Alzheimer’s disease, so the effective identification and early intervention of MCI play an important role in the prevention of Alzheimer’s disease [2].

Mild cognitive impairment can eventually progress to dementia, and while some drug treatments can improve symptoms in people with dementia and Alzheimer’s disease, they do not completely stop the progression of the disease [4]. In addition, the therapeutic effect of these drugs on the above diseases is not significant [5,6]. Therefore, the prevention of and interventions for mild cognitive impairment may be the best ways to prevent the onset of dementia. Interventions for MCI can be divided into the following categories: pharmacological interventions, over-the-counter (OTC) supplements, physical exercise, cognitive interventions, and motor–cognitive interventions. The medical field currently prefers to promote non-pharmacological therapies to avoid causing adverse drug reactions in patients [7,8]. Single cognitive training has been shown to improve targeted cognitive functions, including visuospatial and executive functions and information processing functions. However, the use of dual-task or continuous training, such as motor–cognitive interventions, has been shown to improve cognition more significantly than either type of training alone [9]. There is no clear definition of a motor–cognitive intervention, but some research suggests that a motor–cognitive intervention is a structured intervention that combines cognitive and motor training in the form of dual-task or continuous training [10]. Through cognitive and aerobic exercise training, patients with MCI can strengthen different neural functions, coordinate cognition, play an effective role, and promote the optimization of cognitive function. However, there are still some limitations to this intervention approach. Numerous studies have shown that motor–cognitive interventions can be very effective in maintaining or improving cognitive performance in patients with cognitive impairment. However, it has also been shown that the effect of combined interventions is not advantageous compared with the effect of single-task interventions, mainly because of differences in intervention timing, format, and population variability [11]. Currently, the following problems exist in the study of motor–cognitive interventions: firstly, the optimal form and dosage of training (frequency, intensity, duration) are still uncertain, and there are no uniform standards; secondly, there are no validated experiments on the pre- or post-sequential effects of exercise and cognitive interventions. There is, therefore, a need for more in-depth research on these two aspects, setting up experimental studies with suitable control groups to analyze the optimal dosage and training patterns.

This study aimed to investigate the effects of a motor–cognitive intervention on the cognitive function of older adults with MCI and to design a more appropriate intervention approach for Chinese characteristics. A large number of studies have been conducted on motor–cognitive interventions for MCI, but the variability in diagnostic criteria and MCI definitions, as well as the variability in motor-cognition and inclusion criteria in variability studies, may lead to variability in the effects of motor–cognitive interventions. At the same time, there are few experimental studies on the effectiveness of motor–cognitive interventions for MCI in China and no well-established intervention protocols. This study aims to provide a theoretical basis for the application of motor–cognitive intervention, ensure its application in standards and norms, develop motor–cognitive intervention programs in line with local conditions, and provide a more comprehensive theoretical and practical perspective on motor–cognitive intervention methods. In this way, we can prevent and delay the occurrence of AD, improve the quality of life of older patients with MCI, and reduce the burden on families and society. Based on existing studies, we hypothesize that the intervention group will have significantly improved cognitive abilities compared with the control group.

## 2. Materials and Methods

The present study was a 3-month comparative randomized controlled trial with two groups. We selected a Chinese sample from a regional context and used a self-compiled demographic questionnaire, the Mini-Mental State Examination (MMSE), and the Activity of Daily Living scale (ADLs) as screening instruments. We used a multi-stage cluster random sampling method to select 112 older adults with MCI in Qingdao and divided them into an intervention group and a control group. The intervention group received motor–cognitive intervention training, and the intervention effect was evaluated using the Mini-Mental State Examination (MMSE) and the Montreal Cognitive Assessment (MoCA).

### 2.1. Participants

A multi-stage whole-group random sampling method was used to select 1700 older adults aged 65 years and above in six communities in Qingdao. We used a self-compiled demographic questionnaire, the Mini-Mental State Examination (MMSE) [12], and the Activity of Daily Living scale (ADLs) [11] to screen participants. A total of 1694 participants (M = 72.77 years, SD = 6.47 years, age range = 65 to 94 years) were included, excluding those with incomplete or uncooperative information. This screening identified 239 older adults with MCI, with a prevalence of 14.11%. Considering that the subject group comprises Chinese older adults, this study used the diagnostic criteria from the 2018 Chinese Guidelines for the Treatment of Dementia and Cognitive Impairment [13,14]. The study used a random sampling method to select 112 older adults with MCI as study participants. Based on the random number table method, they were randomly divided into an intervention group and a control group (1:1), with 56 cases in the intervention group and 56 cases in the control group.

Inclusion criteria: (1) age older than 65 years or older; (2) screening results for mild cognitive impairment; (3) inability to walk independently or walking independently with the aid of auxiliary tools; (4) normal communication and language skills (able to understand and correctly answer the researchers’ questions during the screening process); (5) informed consent signed.

Exclusion criteria: (1) cognitive impairment induced by epilepsy, brain trauma, or vascular dementia and diagnosed as dementia; (2) severe organ lesions, such as kidney, heart failure, and other diseases; (3) severe mental illness, such as major depression or schizophrenia; (4) severe impairment of hearing and vision that affects communication; (5) mobility-impaired; (6) illiteracy. The screening results were based on oral reports from the participants and their families and the medical cases they provided.

All participants were informed of the criteria and procedures to ensure informed consent, and all participants volunteered to participate in the study. However, nine participants dropped out because of personal medical reasons or scheduling conflicts. The final analysis was completed with 103 participants (M = 74.45 years, SD = 8.61 years, age range = 65 to 92 years), with 51 in the intervention group (M = 73.41 years, SD = 7.79 years, age range = 65 to 91 years) and 52 in the control group (M = 74.90 years, SD = 6.61 years, age range = 66 to 92 years). The demographic composition of the entire sample and the division of the sample into intervention and control groups are shown in Table 1.

### 2.2. Measures

All participants completed the same questionnaire, and the intervention group completed the intervention plan as follows below.

#### 2.2.1. Demographics

This was a 5-item measure that asked about age, gender, marital status, living arrangement, and education.

#### 2.2.2. Mini-Mental State Examination (MMSE)

The MMSE is a self-report measure that looks at the initial level of cognitive function in older adults with MCI [12]. The MMSE is an 11-item scale that includes temporal orientation, verbal and visuospatial structures, place orientation, short-term memory, immediate memory, attention, and numeracy. The scale is scored out of 30, with higher scores associated with better cognitive functioning. The presence or absence of cognitive impairment is determined based on the literacy level of the respondents. The Shanghai version of the MMSE scale is as follows: (i) Cognitive impairment: The highest score is 30, with scores between 27 and 30 being normal and scores < 27 indicating cognitive impairment. (ii) Classification criteria of dementia at different levels of education: illiterate ≤ 17, primary school ≤ 20, secondary school ≤ 22, college degree or above ≤23. (iii) Dementia severity classification: scores ≥ 21 are mild; scores = 0–20 are moderate; ≤9 is severe [15]. Cronbach’s alpha in this study was 0.778.

#### 2.2.3. Montreal Cognitive Assessment Scale (MoCA)

The MoCA is used to evaluate changes in cognitive function in older adults with MCI before and after intervention [16]. The MoCA consists of 12 questions with 30 individual items, including 7 dimensions such as visuospatial and executive functions, naming, attention, language, abstract thinking skills, delayed recall, and orientation. Each item is scored 1 point for a correct answer and 0 points for an incorrect or no answer. It takes approximately 10 min to complete the test. The total scale score ranges from 0 to 30. In the original version of the scale, 1 point is added for years of education less than or equal to 12 years, with a maximum score of 30 points, and a score of 26 points or more is considered normal. Given the differences in national conditions, the Chinese aged population has a low level of education. For this reason, Chinese scholars have conducted a study on the cut-off value, taking into account the low educational level of the aged population of China, and initially determined a cut-off value of 25 for normal people and MCI patients and an optimal cut-off value of 15 for MCI and AD patients. A score of less than 15 is considered dementia [17]. According to the diagnostic criteria of the 2018 China Dementia and Cognitive Disorders Guidelines [13,14], patients with MCI must have a MoCA score of 15–24. Cronbach’s alpha in this study was 0.798.

#### 2.2.4. Activity of Daily Living Scale (ADLs)

The ADLs allows for the assessment of physical functioning in older people. The ADLs is assessed using the Barthel Index and includes 10 items: eating, bathing, grooming, dressing, bowel control, urinary control, toileting, bed and chair transfer, level walking, and stair walking [17]. The scale is divided into scores of 0, 5, or 10 depending on the level of assistance needed by the older person. The ADLs is scored out of 100. Patients are classified into 4 levels of self-care according to the total score: those with a score of less than 40 are considered to be severely dependent; those with a score of 41–60 are considered to be moderately dependent; those with a score of 61–99 are considered to be mildly dependent; and those with a score of 100 are considered to be fully self-care [18]. Referring to the diagnostic criteria of the 2018 China Dementia and Cognitive Disorders Diagnostic and Treatment Guidelines [13,14], patients with MCI must have an ADLs score greater than or equal to 61. Cronbach’s alpha in this study was 0.924.

#### 2.2.5. Intervention Methods

The experimental group received a uniform intervention at a community health center or a geriatric activity station. The intervention lasted 12 weeks, and the frequency was once every fortnight for 60–90 min each time. After the completion of each intervention, the researchers asked the members of the intervention group about the exercise intensity, and the members answered whether they felt the exercise was difficult or moderate in the form of an oral report. At the end of the intervention, the participants were followed up by the researchers for a total of 12 weeks at a frequency of once every fortnight. The research process as shown in Figure 1.

The intervention consisted of physical movement training and integrated cognitive training. The physical exercise activities were a combination of five senses exercises, brain exercises, and breathing exercises [19,20]. Studies have shown that there are differences in the activation of brain regions during exercise accompanied by rhythm and non-rhythm, and performing repetitive rhythmic exercise may reduce the workload of specific brain regions and release cognitive ability [21]. Several studies have confirmed that music can make movement more rhythmic [22,23,24]. Therefore, to increase the rhythm of the movement and facilitate the participants to follow the rhythm of the music, this study used old songs familiar to the older adults as background music during the exercise. The training was divided into three phases: The first phase is a preparatory phase to warm up and wake up the brain, with participants in a seated position massaging their facial acupuncture points to soft music. The middle phase is aerobic, in which the participants stand and raise their legs alternately, performing brain exercises to fast rhythmic music. In the first section, the five fingers of each hand are tapped in turn; in the second section, the thumbs of both hands are tapped on the remaining fingers in turn; in the third section, the five fingers of both hands are crossed in a fist; in the fourth section, the five fingers of both hands are extended in turn and then retracted in turn. The last stage is the relaxation stage, where participants practice inhaling and exhaling with soft music.

Integrated cognitive training is a comprehensive training method that stimulates multiple domains of cognitive function [25,26]. The aim is to enhance participants’ attention, memory, and executive functions in everyday life. Six interventions were administered in total over a 12-week period. The duration of one intervention was approximately 45 min, and the intervention consisted of the following components: (1) Attention training for 15 min. The aim was to exercise the participants’ degree of control over their attention and improve cognitive performance. For example, the participants were given Schulte squares (25 squares of 1 cm × 1 cm with Arabic numbers from 1 to 25 in any order) and instructed to use their fingers to identify the position of the numbers according to their size and to read them out. (2) Memory training for 15 min. The aim is to improve immediate and delayed memory, auditory memory, and visual memory. ① Auditory memory training: The participants are asked to perform gestures, songs, and number recognition to improve the older adults’ auditory memory function. The content includes gesture listening practice, listening to songs to recognize songs, listening to number addition and subtraction practice, and recognizing different animal sounds; e.g., taking the gesture listening practice as an example, the participants are randomly played gesture commands and asked to make the corresponding gestures quickly. ② Visual memory training: The aim is to improve the ability to remember locations and strengthen the older adults’ ability to remember location, people, etc. The training includes map puzzles, character matching, and simulated trips. Taking a map puzzle as an example, we present the correct map puzzle first and then disrupt the puzzle and ask the participants to put it all together within a specified time. (3) Executive function training for 15 min. The aim is to improve competence in complex activities, learning, planning, and organization. For example, in a simulated supermarket game, participants are asked to shop for items in a designated area, setting different shopping conditions each time, such as type, amount, number of pieces, etc. Participants are asked to follow the shopping conditions. The shopping conditions ranged from simple to difficult. During the training, attention was paid to observing the emotions of the older adults and encouraging them. When shopping, people need to choose specific products in a distracting environment. People with MCI often have some deficits in executive function, which makes it difficult for them to maintain certain purposeful activities, and they are more likely to be distracted than normal people [27]. Therefore, shopping is a challenging skill for MCI patients. In simulated shopping, people need to remember the list of items to be purchased, find out the corresponding product location, and evaluate whether the price of the item is acceptable. In this series of processes, executive function plays a crucial role [28]. Studies have proved that virtual shopping training can be effective in improving executive function and instrumental activities of daily living in patients with MCI [29]. The details of the intervention are shown in Table 2.

Participants in the control group received health education programs of the same content and frequency as those in the intervention group. While the intervention group received the intervention, the control group moved freely in the prescribed place, and no other intervention was added. The control group was asked to avoid cognitive training and exercise training beyond the routine throughout the investigation. Participants in the control group were verbally questioned and evaluated after the health lecture.

### 2.3. Procedure

The content and process of this study complied with the ethical standards of the Qingdao University Ethics Committee and passed a review by the Institutional Review Board of Psychology, Qingdao University. All participants and their families provided informed consent before participation. We anonymized the data using a randomly assigned participant number, which was used throughout the analysis process. All data were used for scientific research only, following the principle of confidentiality.

This study was completed by one graduate student and four community doctors who received homogeneous training. Pre-intervention preparation stage: The intervention plan and route of this project were fully adopted the opinions of neurologists and completed under their guidance; the researchers received unified training to fully understand the goal and significance of this study and precautions in the experiment and to learn and master the methods of motor–cognitive training to ensure the effective implementation of the training. In the intervention implementation stage, the intervention plan was strictly implemented, and relevant records were made. The researchers organized weekly meetings to discuss the problems in the study and formulate corresponding measures to solve them.

The study lasted from July 2020 to March 2021. The researchers called the participants to the community health center and used verbal questioning to help them complete the data collection process. Participants were again assessed with the Montreal Cognitive Assessment Scale before the start of the intervention at the end of the 3-month intervention and at the end of the 3-month follow-up. Whereas participants and physicians allocated to the intervention group were aware of the allocated arm, outcome assessors and data analysts were kept blinded to the allocation.

### 2.4. Sample Size Calculation

The primary screening sample size calculation referred to the previous literature [30], and the calculation formula was as follows:(1)$$n=\frac{u_{\alpha /2}^{2}p(1-p)}{\delta^{2}}$$
where *n* is the sample size; *u_α_*_/2_ is the *u*-value when the cumulative probability is equal to *α*/2 in the normal distribution; and *p* is the estimated prevalence of MCI. According to previous relevant studies, the prevalence of MCI in older adults in the Qingdao community is 20.11% [31]; with a margin of error of *δ* = 2% and *α* = 0.05, the sample size was 1543. The sample size was increased by 10% after taking into account the rate of loss to follow-up, questionnaire efficiency, and systematic error, and the final sample size was determined to be 1700. The formal experimental sample size calculation process is shown in Figure 2.

### 2.5. Data Analysis

Data were analyzed with the use of SPSS 26.0 by investigators who were not involved in guiding the intervention. According to whether the data had a normal distribution, the measurement data were described by frequency and standard deviation or as median and quartile and compared by two independent sample *t*-tests or the Mann–Whitney U rank sum test. Count data were described by their frequency and percentage and compared with the chi-square test. Repeated measures ANOVA was used to explore trends in MoCA and MMSE scores in the two study groups before (T1), at the completion of the intervention (T2), and 3 months after the end of the intervention (T3). In the above tests, *p* < 0.05 was considered statistically significant. The primary outcome of the study was an improvement between the baseline and the 12-week follow-up in cognitive performance, as measured by MoCA and MMSE.

## 3. Results

### 3.1. Comparison of the Demographic Characteristics of the Two Groups

Of the 103 participants in the sample, 67 were female (65.05%), and 34.95% were male. The demographic characteristics of the intervention and control groups are listed in Table 1. For gender, marital status, residential status, and education, statistical analyses were conducted using chi-square tests, and for age, independent samples *t*-tests were used to compare the intervention and control groups (see Table 1). The results revealed no significant differences between the two groups of MCI older adults in terms of gender, age, marital status, residential status, or education level (*p* > 0.05). This indicated that the general demographic information of the two groups of MCI older adults was comparable.

### 3.2. Comparison of Initial Daily Living Ability between the Two Groups

The Shapiro–Wilk test showed that the ADL scores of the two groups were not normally distributed. Therefore, we used the Mann–Whitney U rank sum test to compare the differences in the activities of daily living between the two groups of subjects. The participants in the two groups were not statistically different in the initial level of activities of daily living (*p* > 0.05), which was balanced and comparable (see Table 1).

### 3.3. Analysis of the Effect of the Intervention

#### Comparison of the Total MoCA and MMSE Scores of the Two Groups

The total MoCA and MMSE scores followed a normal distribution in the S-W test. Therefore, in this study, a repeated measures ANOVA was performed on the total MoCA and MMSE scores before the intervention (T1), at the completion of three months of the intervention (T2), and three months after the intervention (T3) using independent samples *t*-tests to compare the two groups at each time point and the Bonferroni method to make a two-by-two comparison of the three time points for each group. The results showed that there was no significant difference in total MoCA and MMSE scores between the two groups (*p* > 0.05), indicating that the initial cognitive level of the two groups was not significantly different, and the results were comparable. For the results, see Table 3.

The results of the repeated measures ANOVA on the total MoCA score revealed that the main effect of time was significant (*F* = 28.081, *p* < 0.001, η^2^_P_ = 0.360), the main effect of the group was significant (*F* = 30.277, *p* < 0.001, η^2^_P_ = 0.231), and the interaction effect was significant (*F* = 39.550, *p* < 0.001, η^2^_P_ = 0.442). The simple effect results for the time showed that a simple effect for the group was not significant at T1, *F* = 0.023, *p* > 0.05, η^2^_P_ < 0.001; a simple effect for the group was significant at T2, *F* = 58.336, *p* < 0.001, η^2^_P_ = 0.366; and a simple effect for the group was significant at T3, *F* = 58.209, *p* < 0.001, η^2^_P_ = 0.366. The group simple effects results showed the non-significant simple effect of time in the control condition (*F* = 1.171, *p* > 0.05, η^2^_P_ = 0.023) and the significant simple effect of time in the intervention condition (*F* = 65.833, *p* < 0.001, η^2^_P_ = 0.568). Multiple comparisons revealed no significant difference in scores for the control group at the three time points (*p* > 0.05). For the intervention group at T1, the scores were significantly lower than the T2 and T3 scores (*p* < 0.001), and there was no significant difference between the T2 and T3 scores (*p* > 0.05). The *t*-test results showed that there was no significant difference between the two groups’ scores in this dimension at T1 (*p* > 0.05) and a significant difference between the two groups’ scores in this dimension at T2 and T3 (*p* < 0.001). The overall cognitive level of the MCI older adults in the intervention group tended to increase over time, with a small but insignificant decrease three months after the end of the intervention; in the control group, the cognitive level of the MCI older adults remained largely stable; see Figure 3. We used one-way repeated measures analysis of variance to analyze MoCA scores in the intervention group; the results showed that the MoCA scores differed significantly across the three time points in the intervention group (*F* = 59.646, *p* < 0.001, η^2^_P_ = 0.544).

The results of the repeated measures ANOVA for the total MMSE score revealed that the main effect of time was significant (*F* = 9.821, *p* < 0.001, η^2^_P_ = 0.164), the main effect of the group was significant (*F* = 25.587, *p* < 0.001, η^2^_P_ = 0.202), and the interaction effect was significant (*F* = 24.614, *p* < 0.001, η^2^_P_ = 0.330). The simple effect results for the time showed that a simple effect for the group was not significant at T1, *F* = 0.934, *p* > 0.05, η^2^_P_ = 0.009; a simple effect for the group was significant at T2, *F* = 43.952, *p* < 0.001, η^2^_P_ = 0.303; and a simple effect for the group was significant at T3, *F* = 40.168, *p* < 0.001, η^2^_P_ = 0.285. The group simple effects results showed the non-significant simple effect of time in the control condition (*F* = 2.059, *p* > 0.05, η^2^_P_ = 0.040) and the significant simple effect of time in the intervention condition (*F* = 32.085, *p* < 0.001, η^2^_P_ = 0.391). Multiple comparisons revealed no significant difference in scores for the control group at the three time points (*p* > 0.05). For the intervention group at T1, the scores were significantly lower than the T2 and T3 scores (*p* < 0.001), and there was no significant difference between the T2 and T3 scores (*p* > 0.05). The *t*-test results showed that there was no significant difference between the two groups’ scores on this dimension at T1 (*p* > 0.05) and a significant difference between the two groups’ scores on this dimension at T2 and T3 (*p* < 0.001). The overall cognitive level of the MCI older adults in the intervention group tended to increase over time and remained slightly elevated for three months after the end of the intervention, without reaching significant levels; in the control group, the cognitive level of the MCI older adults remained largely stable; see Figure 3. We used one-way repeated measures analysis of variance to analyze MMSE scores in the intervention group; the results showed that MMSE scores differed significantly across the three time points in the intervention group (*F* = 32.517, *p* < 0.001, η^2^_P_ = 0.394).

In order to gain a deeper understanding of which aspects of cognitive functioning were brought about by the intervention training methods used in this study, the MoCA sub-scores of the intervention and control groups were next analyzed and compared to explain the possible effects of the training. The data for each group obeyed a normal distribution by S-W test. Therefore, in this study, a repeated measures ANOVA was conducted on the MoCA sub-scores before the intervention (T1), at the completion of three months of the intervention (T2), and three months after the intervention (T3); the two groups were compared at each time point using an independent samples *t*-test, and two-by-two comparisons were made at three time points for each group using the Bonferroni method. For the results, see Table 4.

In the visuospatial/executive dimension, the main effect of time was significant (*F* = 2.010, *p* < 0.001, η^2^_P_ = 0.176), the group main effect was not significant (*F* = 1.945, *p* > 0.05, η^2^_P_ = 0.019), and the interaction effect was significant (*F* = 8.510, *p* < 0.001, η^2^_P_ = 0.145). The results of the simple effect of time showed that the simple effects for the group at T1 were not significant (*F* = 0.428, *p* > 0.05, η^2^_P_ = 0.004), the group simple effects were significant at T2 (*F* = 5.505, *p* < 0.05, η^2^_P_ = 0.052), and the group simple effects were significant at T3 (*F* = 5.837, *p* < 0.05, η^2^_P_ = 0.055). The group simple effects results showed that the simple effect for the control condition was not significant (*F* = 0.798, *p* > 0.05, η^2^_P_ = 0.160), and the simple effect for time in the intervention condition was significant (*F* = 18.205, *p* < 0.001, η^2^_P_ = 0.267). Multiple comparisons revealed no significant difference between the control group scores at the three time points (*p* > 0.05); the intervention group scored significantly lower at T1 than at T2 and T3 (*p* < 0.001), with no significant difference between the T2 and T3 scores (*p* > 0.05). The results of the *t*-test showed that there was no significant difference between the scores of the two groups in this dimension at T1 (*p* > 0.05) and a significant difference between the scores of the two groups in this dimension at T2 and T3 (*p* < 0.05). There was no significant difference in scores in this dimension between the two groups before the intervention began. Over time, the visuospatial and executive abilities of MCI older adults in the intervention group showed a significant upward trend and remained slightly elevated for three months after the end of the intervention, without reaching significant levels; in the control group, the visuospatial and executive abilities of MCI older adults did not change significantly; see Figure 3.

In the naming dimension, the main effect of time was significant (*F* = 12.689, *p* < 0.001, η^2^_P_ = 0.202), the main effect of group was not significant (*F* = 1.284, *p* > 0.05, η^2^_P_ = 0.013), and the interaction effect was significant (*F* = 10.250, *p* < 0.001, η^2^_P_ = 0.170). The results for the simple effect of time showed that the group simple effect at T1 was not significant (*F* = 1.058, *p* > 0.05, η^2^_P_ = 0.010), the group simple effect at T2 was significant (*F* = 5.520, *p* < 0.05, η^2^_P_ = 0.010), and the group simple effect at T3 was significant (*F* = 5.873, *p* < 0.05, η^2^_P_ = 0.055). The simple effect results for the group showed the non-significant simple effect of time in the control condition (*F* = 1.288, *p* > 0.05, η^2^_P_ = 0.025) and the significant simple effect of time in the intervention condition (*F* = 21.455, *p* < 0.001, η^2^_P_ = 0.300). Multiple comparisons revealed no significant differences between the control group scores at the three time points (*p* > 0.05); the intervention group scores were significantly lower at T1 than at the latter two time points (*p* < 0.001), and there were no significant differences between T2 and T3 scores (*p* > 0.05). The results of the *t*-test showed that there was no significant difference between the scores of the two groups in this dimension at T1 (*p* > 0.05) and a significant difference between the scores of the two groups in this dimension at T2 and T3 (*p* < 0.05). There was no significant difference in naming ability between the two groups at T1. Over time, the naming ability of the intervention group increased significantly and remained slightly elevated for three months after the end of the intervention, but the difference was not significant; the overall change in the naming ability of the control group was not significant; see Figure 3.

In the attention dimension, the main effect of time was significant (*F* = 3.680, *p* < 0.05, η^2^_P_ = 0.069), the main effect of group was significant (*F* = 4.318, *p* < 0.05, η^2^_P_ = 0.041), and the interaction effect was not significant (*F* = 0.860, *p* > 0.05, η^2^_P_ = 0.017). The results of the main effects analysis for time showed that the difference between T1 and the latter two time points in this dimension score was not significant (*p* > 0.05), and the difference between T2 and T3 was significant (*p* < 0.05). The results of the group main effects analysis showed that there was a significant difference between the intervention and control groups (*p* < 0.05), with the intervention group scoring significantly higher than the control group. The *t*-test showed that there was no significant difference between the two groups in this dimension at T1 (*p* > 0.05), and there was a significant difference between the two groups at T2 and T3 (*p* < 0.05).

In the language dimension, the main effect of time was significant (*F* = 4.336, *p* < 0.05, η^2^_P_ = 0.080), the main effect of group was not significant (*F* = 3.284, *p* > 0.05, η^2^_P_ = 0.031), and the interaction effect was significant (*F* = 13.324, *p* < 0.001, η^2^_P_ = 0.210). The results for the simple effect of time showed that the group simple effect at T1 was significant (*F* = 5.586, *p* < 0.05, η^2^_P_ = 0.052), the group simple effect at T2 was significant (*F* = 11.940, *p* = 0.001, η^2^_P_ = 0.106), and the group simple effect at T3 was significant (*F* = 12.175, *p* = 0.001, η^2^_P_ = 0.108). The simple effect results for the group showed that the simple effect of time was significant in the control condition (*F* = 6.340, *p* < 0.01, η^2^_P_ = 0.113) and in the intervention condition (*F* = 11.271, *p* < 0.001, η^2^_P_ = 0.184). Multiple comparisons revealed no significant difference between the control group scores at the three time points (*p* > 0.05); the intervention group scored significantly lower at T1 than at T2 and T3 (*p* < 0.001) and showed no significant difference between the T2 and T3 scores (*p* > 0.05). The results of the *t*-test showed that the scores in this dimension were significantly different between the two groups at all three time points (*p* < 0.05), with the language ability of the control group being significantly higher than that of the control group before the intervention, possibly because our screening criteria did not place certain restrictions on the educational level of the participants. However, after starting the intervention, the intervention group showed a rapid and significant improvement in language skills and overtook the control group and then showed a slow but non-significant decline; the control group showed a consistent downward trend in language skills; see Figure 3.

In the abstraction dimension, the main effect of time was significant (*F* = 4.726, *p* < 0.05, η^2^_P_ = 0.176), the main effect of group was significant (*F* = 3.988, *p* < 0.05, η^2^_P_ = 0.038), and the interaction effect was significant (*F* = 4.726, *p* < 0.05, η^2^_P_ = 0.176). The results for the simple effect of time showed that the group simple effect at T1 was not significant (*F* = 0.008, *p* > 0.05, η^2^_P_ < 0.001), the group simple effect at T2 was significant (*F* = 5.247, *p* < 0.05, η^2^_P_ = 0.049), and the group simple effect at T3 was significant (*F* = 10.604, *p* < 0.01, η^2^_P_ = 0.095). The simple effect results for the group showed a significant simple effect for time in the control condition (*F* = 5.319, *p* < 0.01, η^2^_P_ = 0.096), and the simple effect of time was significant in the intervention condition (*F* = 4.144, *p* < 0.05, η^2^_P_ = 0.077). Multiple comparisons revealed no significant difference between T1 scores and both T2 and T3 in the control group (*p* > 0.05) and showed no significant difference between T2 and T3 (*p* > 0.05); the intervention group scored significantly lower at T1 than at T2 and T3 (*p* < 0.05) and showed no significant difference between the T2 and T3 scores (*p* > 0.05). The results of the *t*-test showed that there was no significant difference between the two groups’ scores in this dimension at T1 (*p* > 0.05) and a significant difference at T2 and T3 (*p* < 0.05). There was no significant difference in abstraction ability between the two groups at T1, with scores in the intervention group improving significantly over time and remaining essentially flat three months after the intervention; abstraction ability in the control group declined significantly in the later stages; see Figure 3.

In the delayed recall dimension, the main effect of time was significant (*F* = 16.319, *p* < 0.001, η^2^_P_ = 0.246), the group main effect was significant (*F* = 6.309, *p* < 0.05, η^2^_P_ = 0.059), and the interaction effect was significant (*F* = 14.549, *p* < 0.001, η^2^_P_ = 0.225). The results for the simple effect of time showed that the group simple effect at T1 was not significant (*F* = 1.834, *p* > 0.05, η^2^_P_ = 0.018), the group simple effect at T2 was significant (*F* = 16.245, *p* < 0.001, η^2^_P_ = 0.139), and the group simple effect at T3 was significant (*F* = 9.960, *p* < 0.01, η^2^_P_ = 0.090). The simple effect results for the group showed that the simple effect of time in the control condition was not significant (*F* = 0.202, *p* > 0.05, η^2^_P_ = 0.004), and in the intervention condition, it was significant (*F* = 30.373, *p* < 0.001, η^2^_P_ = 0.378). Multiple comparisons revealed no significant differences between the control group scores at the three time points (*p* > 0.05) and significant differences between the intervention group scores at T1, T2, and T3 (*p* < 0.001). The *t*-test showed no significant difference between the two groups’ scores in this dimension at T1 (*p* > 0.05) and a significant difference at T2 and T3 (*p* < 0.05). There was no significant difference in the level of delayed recall between the two groups before the intervention, with a significant increase in the intervention group’s delayed recall ability over time and a significant downward trend in this ability three months after the intervention was completed; there was no significant change in the control group; see Figure 3.

In the orientation dimension, the main effect of time was not significant (*F* = 2.221, *p* > 0.05, η^2^_P_ = 0.043), the group main effect was significant (*F* = 4.895, *p* < 0.05, η^2^_P_ = 0.046), and the interaction effect was not significant (*F* = 0.862, *p* > 0.05, η^2^_P_ = 0.017). The results of the group main effects analysis showed that the intervention and control groups differed significantly in their scores in this dimension (*p* < 0.05), with the intervention group scoring significantly higher than the control group. The *t*-test showed that there was no significant difference between the two groups in this dimension at T1 (*p* > 0.05) and a significant difference at T2 and T3 (*p* < 0.05).

## 4. Discussion

This study investigated whether a motor–cognitive intervention could be used as an intervention to improve cognitive function in patients with MCI in the community using a combination of physical movement activities and integrated cognitive training. A total of 103 participants were included in the data analysis. The results of the data analysis showed that, after 3 months of intervention and 3 months of follow-up, the cognitive function of MCI older adults improved significantly to some extent, largely in line with our expected hypothesis, which is also consistent with the results of previous studies [32,33].

### 4.1. Analysis of General Information for the Intervention and Control Groups

A total of 103 community-based MCI participants were included in this study, predominantly female (a male-to-female ratio of 1:1.86), with a generally low level of education, and most of the study participants had a spouse or lived with a spouse or family member. The main reason for the large number of female participants may be that the older female population is more receptive to brain fitness exercises, and older men are more inclined to participate in strength-based exercises and, therefore, less willing to participate in this training [34]. Eighty-three of the study participants had a partner, accounting for 81.59% of the total. Ninety-six seniors lived with a partner or family member, accounting for 93.20% of the total, which may be related to the fact that the majority of the study participants in this study were senior citizens. The general low educational level of the older patients with MCI included in this study, with 39 (37.86% of the total) having a junior high school education or above, may be due to the generally low educational level of older adults in the community where the participants lived.

### 4.2. Analysis of the Implementation of the Intervention Plan

The intervention protocol of this experiment comprehensively combines the motor ability and cognitive level of MCI patients in the community. By referring to the relevant research at home and abroad, the intervention program was revised by neurological experts from the scientific nature and applicability of the program, and five senses exercises, brain exercises, and breathing training with Chinese characteristics were used as the exercise–cognitive intervention program in this study. During the intervention, a total of nine people were lost, including five in the intervention group and four in the control group, with a loss rate of 8.04%. This indicates that the older people in the intervention group had good compliance and showed great interest in the intervention plan and that there were no intervention-related adverse events during the intervention. At the end of each intervention, verbal reports of exercise intensity made by the members of the intervention group were moderate, and no members felt fatigued by exercise training. All of these results suggest that the motor–cognitive intervention plan in this study is suitable for older people with MCI and is economical, simple, fun, and easy for older people to master.

### 4.3. Intervention Effects of the Exercise-Cognitive Intervention

#### 4.3.1. Improving Cognitive Function

The results showed a significant upward trend in the total MoCA score over time in the intervention group compared with the control group, with a significant difference in the total MoCA score between the two groups (*p* < 0.05), which is consistent with previous studies. Many studies [35,36] have shown that aerobic exercise can strengthen fitness ability and improve brain function and structure. Cognitive training can selectively improve brain function, mainly improving memory, visuospatial abilities, attention, balance, and executive abilities in MCI patients. Moderate-intensity aerobic exercise can effectively improve cognitive function in older patients with MCI [37]. Exercise therapy has the advantage of being simple to implement and easy to learn, which helps older people to accept and adhere to it [38]. Related animal studies have explored the rationale for the effects of motor–cognitive interventions on cognitive function [39], suggesting that combining exercise and cognitive training can accelerate neurogenesis and vascular regeneration, which can be more effective in improving cognition than training in individual domains [40]. Hagovska et al. [41] used older adults with MCI as the study target and instructed them to undergo cognitive and motor training. The duration of the trial was 10 weeks, allowing them to receive the intervention three times a week. Upon completion of the intervention, the study participants in the intervention group had significantly higher MMSE assessment scores, balance test scores, and walking function test scores than the control group, and their ADLs values were significantly improved. The interventions used in this study may have had some effect in enhancing the patients’ visuospatial, naming, language, attention, abstraction skill, and delayed recall levels. However, there may be some degree of deficiency in the training of orientation skills; the intervention training also brought some improvement in orientation skills in older adults with MCI, but the effect was not significant. This implies that the intervention using a combination of exercise and cognitive training for MCI patients in this study may have contributed to some level of improvement in cognitive functions in this group.

In previous studies, a single-domain approach to cognitive function training has mostly been used [42,43]. However, studies have shown that mental activity training or physical training alone has a low impact on specific domains of influence and cognitive functioning in older people, while combined interventions can generate synergistic effects [44]. Comprehensive cognitive training is more effective than individual-domain cognitive training, with gains in reasoning being maintained for up to one year [45]. Feng et al. [46] targeted 90 community-dwelling older adults over the age of 70 in a trial of 24 sessions in a comprehensive cognitive intervention over a 12-week period. The results showed that community-dwelling older adults could effectively improve or maintain their reasoning and executive abilities, with a particularly significant improvement in reasoning. The results showed that the intervention training effects in this study were better maintained at the level of three dimensions: visuospatial and executive, naming, and abstraction. Three months after the end of the intervention, the levels of these three dimensions were still increasing slightly or remained stable in the intervention group. This indicates that the integrated cognitive training approach used in this study was effective in maintaining the improvement in visuospatial and executive functions, naming, and abstraction. However, three months after the end of the intervention, the intervention group’s levels in the attention, language, and delayed recall dimensions decreased, with significant decreases in attention and delayed recall. It is, therefore, hypothesized that the intervention training method in this study has some limitations on the maintenance of these three dimensions, which provides a direction for the further improvement of the intervention in the future.

#### 4.3.2. Trends in Cognitive Function

The results of this study pointed out that the trends in the cognitive level of the two groups of MCI patients were different. The analysis of the total MoCA score showed that the overall cognitive level of the intervention group increased significantly within three months of starting the intervention and that the increase in cognitive ability showed a positive correlation with the length of the intervention. At present, there are few studies on changes in the time course of cognitive benefits produced by motor–cognitive combined intervention modes. Zhu et al. [47] conducted a meta-analysis and showed that only a longer intervention time (12 weeks) was more effective, and the optimal training time, frequency, and intensity still need to be more accurately evaluated. Previous meta-analyses have also suggested that longer intervention durations are not necessary to produce cognitive effects in older adults in studies of exercise interventions [45,48]. At the time of the completion of the intervention and follow-up, the study participants were influenced by family and environmental factors and did not undertake the exercise and cognitive training as planned. The results of the study showed that the overall cognitive level of the intervention group decreased to a certain extent after 3 months of the intervention compared with a tendency to increase during the intervention. This also suggests that there is a time limit to the improvement and maintenance of cognitive levels caused by the intervention. Overall, the motor–cognitive intervention was effective in improving cognition in the aged MCI population, but there was a “dose effect”, which refers to the relationships between the components of exercise, such as type, duration, intensity, and frequency, individually and interactively, and executive function [49], meaning that motor–cognitive training shows a positive relationship with health benefit factors [50]. Alternatively, sample characteristics (gender, cognitive impairment, baseline performance) and training parameters (intensity, frequency, duration) have been shown to potentially modify cognitive benefits after exercise–cognitive interventions [51].

This study showed that the longer the duration and higher the completion of the motor–cognitive training performed by the study participants, the more significant the improvement in cognitive level. Compared with the intervention group, the cognitive levels of the control group of older people with MCI changed less across time, suggesting that the current level of cognition and self-management of older people with MCI provides less motivation to engage in physical activity, cognitive training, and participation in social activities and that guidance and assistance from medical professionals is still required.

#### 4.3.3. Limitations and Future Research

In this study, the MoCA and MMSE scales used in most previous research were used to evaluate the cognitive ability of older adults with MCI, which is representative of MCI research. As research in this field continues to deepen, we need to further refine and enrich the findings to cover a more complete neuropsychological assessment of different cognitive domains. For example, Marta Bisbe [52] used a comprehensive, standardized neuropsychological test to compare the cognitive effects of dance exercises and multimodal physical therapy in older patients with the Word List Learning test from the Wechsler Memory Scale—Third Edition (WMS-III), the visual memory subtest of the Repeatable Battery for the Assessment of Neuropsychological Status (RBANS), the Trail Making Test (A and B), Letter Verbal Fluency (LVF) and Category Verbal Fluency (CVF), the Boston Naming Test (BNT), and Judgment of Line Orientation. Future studies should not only increase the evaluation of physiological indicators, but also use more objective biological indicators and imaging examinations to comprehensively understand the intervention effect. Future studies should try to explore the mechanism of MCI related risk factors and determine the specific cognitive impairment from the fields of neuroelectrophysiology, imaging, and molecular biology. By incorporating data from multiple sources for analysis and developing intervention approaches with different focuses based on closely related risk factors, we can effectively alleviate cognitive impairment in different patients and provide a basis for early intervention in Alzheimer’s disease.

The intervention training program in this study was based on the agreeability of most older adults, but older MCI patients are heterogeneous, including the acceptance of exercise intensity. For future studies, we could adopt quantifiable evaluation metrics for exercise intensity, such as using the percentage reserve heart rate method to determine the target heart rate range and using relevant equipment to obtain the participants’ heart rate METS and oxygen consumption during exercise. The accurate acquisition of parameters, such as participants’ motor abilities, will help us to further develop more targeted training content and intensity according to individual differences in older adults with MCI.

The single sampling source of this study makes it difficult to compare the differences between MCI patients in different regions and institutions. Therefore, it is more helpful to include patients from different regions, hospitals, and nursing homes to compare the internal differences of MCI patients in different regions of China in order to explore the effect of motor–cognitive intervention in these different regions. At the same time, in order to obtain more stable and generalizable experimental results, it is necessary to expand the sample size. Secondly, given the large number of samples in the initial screening and the desire to follow the rules of randomization as much as possible, we did not control the type, degree, or cause of cognitive impairment in the older adults when selecting the MCI sample. Also, because our sampling was in the community and not in a standard healthcare facility, the heterogeneity between the screened patients was large compared with that in a healthcare facility, so we did not assess MCI patients precisely, which is an important issue that is difficult to reconcile with large sample randomization studies in a community setting. Although clinic-based longitudinal studies of MCI are less unstable than population-based studies, and population-based studies using less precise clinical diagnostic tools may show more annual differences, these differences will eventually diminish with longitudinal follow-ups [53]. To compensate for the instability of the experimental results, caused by randomized sampling in community, we can use longitudinal designs and long-term follow-up studies in future research.

## 5. Conclusions

Alzheimer’s disease is the leading cause of disability and incapacity in older people worldwide. It has a significant impact not only on individuals but also on families, communities, and society. The use of scientific screening methods and effective interventions can significantly reduce the incidence and prevalence of Alzheimer’s disease, improve the quality of life of older people, and promote family harmony. This study designed a motor–cognitive intervention that consisted of a combination of five senses exercises, brain exercises, and breathing exercises, as well as an integrated training approach that stimulated multiple areas of cognitive function, providing a theoretical basis for the application of motor–cognitive intervention modalities. The results of the study suggest that a motor–cognitive intervention may be able to improve the cognitive function of older adults with MCI to some extent, which supports the findings of previous studies. This method is simple, easy to implement, and fun and has shown high acceptance and good compliance among older patients with MCI, which might be an effective method of intervention for older patients with MCI and could be further promoted in this group.

## Figures and Tables

**Figure 1 behavsci-13-00737-f001:**
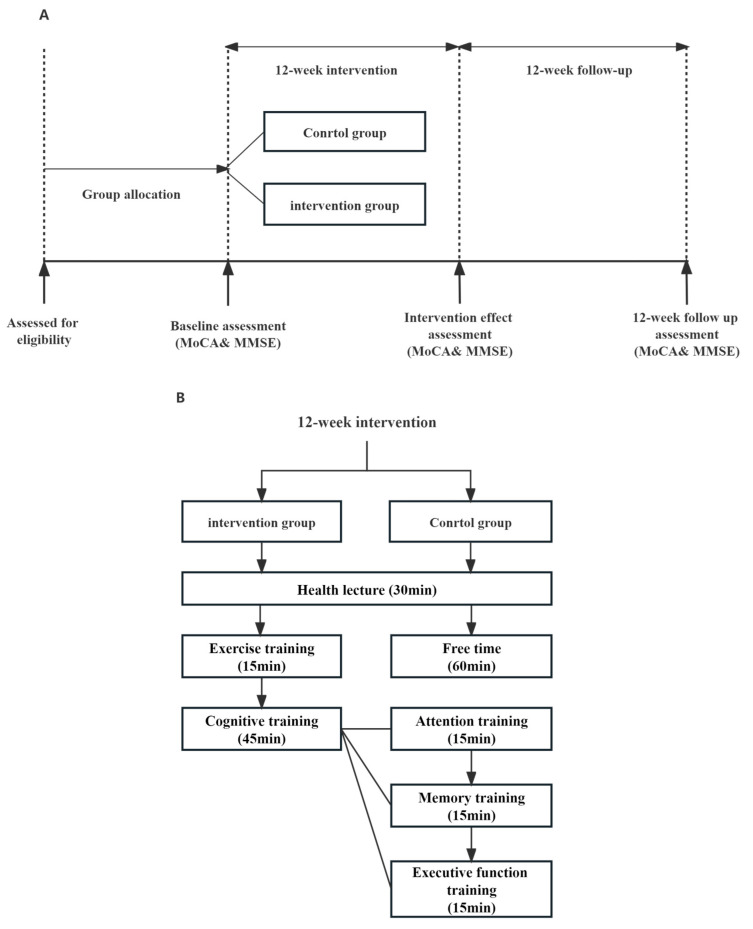
Flow diagram of the intervention. (**A**) The overall study process was divided into three parts: grouping participants and collecting their baseline information; A 12-week intervention experiment was conducted, and data were collected again at the end of the experiment; Follow-up was conducted for 12 weeks, with the final data collection at the completion. (**B**) 12 weeks intervention process. The intervention and control groups received consistent health lectures in separate rooms in the same facility. After the lecture, the intervention group performed 15 min of motor training followed by 45 min of cognitive training; The control group had 60 min of free activity.

**Figure 2 behavsci-13-00737-f002:**
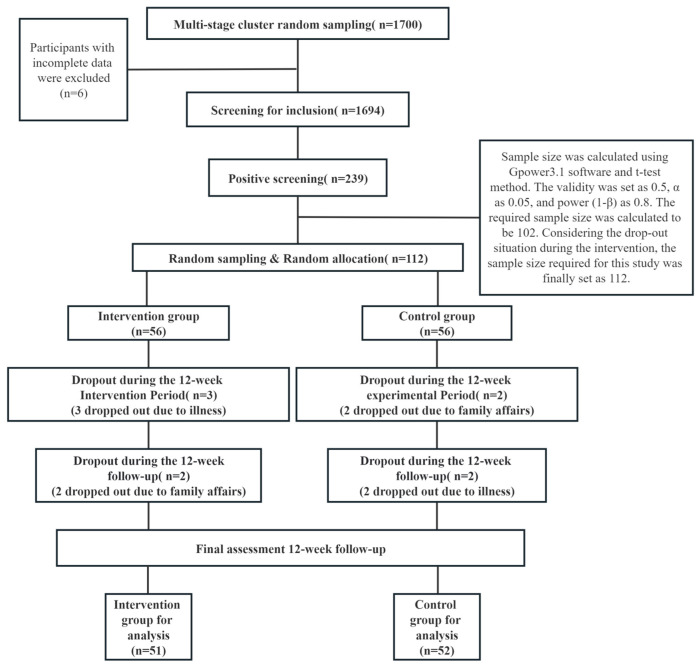
Flow diagram from initial contact to study completion.

**Figure 3 behavsci-13-00737-f003:**
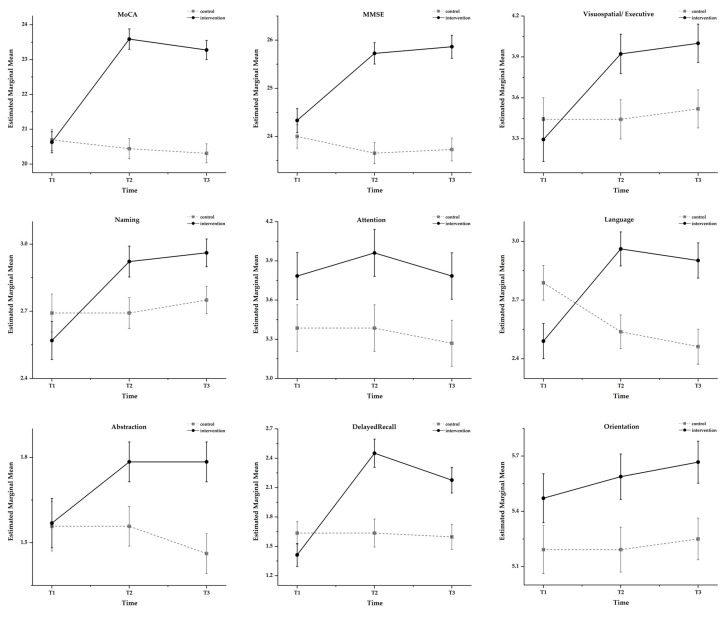
Comparison of changes in cognitive levels between the two groups.

**Table 1 behavsci-13-00737-t001:** Comparison of general demographic information between the intervention and control groups (*n* = 103).

Variables	Classification	Intervention Group (*n* = 51)	Control Group (*n* = 52)	*t/χ^2^/Z*	*p*
Age		73.41 ± 7.785	74.90 ± 6.613	1.05 ^a^	0.297
Gender	Male	16 (31.37%)	20 (38.46%)	0.341 ^b^	0.560
	Female	35 (68.63%)	32 (61.54%)		
Marital status	No partner	8 (15.69%)	12 (23.08%)	0.840 ^b^	0.361
	With partner	43 (84.31%)	40 (76.92%)		
Living arrangement	Living alone	3 (5.88%)	4 (7.69%)	0.000 ^b^	1.000
	Not living alone	48 (94.12%)	48 (92.31%)		
Education	Below junior high school	29 (56.86%)	35 (67.31%)	2.448 ^b^	0.121
	Lower secondary and above	22 (43.14%)	17 (32.69%)		
ADL	\	100.00 (100.00,100.00)	100.00 (95.00,100.00)	1.544 ^c^	0.123

^a^ Independent samples *t*-test, ^b^ chi-square test, ^c^ Mann–Whitney U rank sum test.

**Table 2 behavsci-13-00737-t002:** Components of motor–cognitive intervention training.

	Topics	Contents
First	Getting to know each other and gaining trust	Describing the purpose of the intervention, the members of the team
		Group members introduce themselves to each other
	A proper understanding of MCI disease and dementia	Explaining the relationship between MCI disease and dementia
	Sports interventions	Five senses exercise, brain exercise (aerobic exercise), breathing exercise
	Cognitive interventions	Attention training: Schulte squares
		Memory training: gestures practice; map puzzles
		Executive function training: simulated shopping training
Second	Relationship: cognitive impairment and mental health	What is mental health and depression?
	Sports interventions	Five senses exercise, brain exercise, breathing exercise
	Cognitive interventions	Attention training: Schulte squares
		Memory training: song recognition; character matching
		Executive function training: simulated shopping training
Third	Relationship: cognitive impairment and movement	Choosing appropriate exercises and developing an exercise plan
	Sports interventions	Five senses exercise, brain exercise, breathing exercise
	Cognitive interventions	Attention training: Schulte squares
		Memory training: number addition and subtraction; simulated trips
		Executive function training: simulated shopping training
Fourth	Relationship: cognitive impairment and sleep	Introduction to the importance of sleep
		How to improve the quality of your sleep
		Factors causing sleep disorders
	Sports interventions	Five senses exercise, brain exercise, breathing exercise
	Cognitive interventions	Attention training: Schulte squares
		Memory training: animal sound recognition; map puzzles
		Executive function training: simulated shopping training
Fifth	Relationship: cognitive impairment and chronic illness	Types of chronic diseases
		Risk factors for chronic diseases
		The dangers of chronic disease for MCI disease
	Sports interventions	Five senses exercise, brain exercise, breathing exercise
	Cognitive interventions	Attention training: Schulte squares
		Memory training: gestures practice; character matching
		Executive function training: simulated shopping training
Sixth	Experience	Effective exercise–cognitive training techniques
	Sports interventions	Five senses exercise, brain exercise, breathing exercise
	Cognitive interventions	Attention training: Schulte squares
		Memory training: song recognition; simulated trips
		Executive function training: simulated shopping training

**Table 3 behavsci-13-00737-t003:** Total MoCA scores were compared between the two groups (*n* = 103).

	T1	T2	T3	Repeat Measurement F-Test	
	M ± SD			*F*	η^2^_P_
MoCA					
intervention group	20.63 ± 2.088	23.59 ± 1.899	23.27 ± 1.799		
control group	20.69 ± 2.280	20.44 ± 2.261	20.31 ± 2.147		
*t*	0.151	−7.651	−7.643		
*p*	0.881	0.000	0.000		
Group				30.277 ***	0.231
Time				28.081 ***	0.36
Group × Time				39.550 ***	0.442
MMSE					
intervention group	24.33 ± 1.785	25.73 ± 1.343	25.86 ± 1.456		
control group	24.00 ± 1.715	23.65 ± 1.792	23.73 ± 1.921		
*t*	−0.966	−6.648	−6.355		
*p*	0.336	0.000	0.000		
Group				25.587 ***	0.202
Time				9.821 ***	0.164
Group × Time				24.614 ***	0.33

*** *p* < 0.001.

**Table 4 behavsci-13-00737-t004:** MoCA sub-scores were compared between the two groups (*n* = 103).

		T1			T2			T3			Time	Group	Time × Group
	*n*	Mean ± SD	*t*	*p*	Mean ± SD	*t*	*p*	Mean ± SD	*t*	*p*	*F*		
Visuospatial/Executive													
Intervention group	51	3.29 ± 1.101	0.655	0.514	3.92 ± 0.845	−2.354	0.021 *	4.00 ± 0.894	−2.421	0.017 *	10.662 ***	1.945	8.510 ***
Control group	52	3.44 ± 1.195			3.44 ± 1.195			3.52 ± 1.111					
Naming													
Intervention group	51	2.57 ± 0.575	1.029	0.306	2.92 ± 0.272	−2.366	0.021 *	2.96 ± 0.196	−2.443	0.017 *	12.689 ***	1.284	10.250 ***
Control group	52	2.69 ± 0.643			2.69 ± 0.643			2.75 ± 0.590					
Attention													
Intervention group	51	3.78 ± 1.064	−1.582	0.117	3.96 ± 1.058	−2.286	0.025 *	3.78 ± 1.119	−2.063	0.042 *	3.680 *	4.318 *	0.86
Control group	52	3.38 ± 1.471			3.38 ± 1.471			3.27 ± 1.402					
Language													
Intervention group	51	2.49 ± 0.674	2.361	0.02 *	2.96 ± 0.196	−3.486	0.001 ***	2.90 ± 0.300	−3.516	0.001 ***	4.336 *	3.284	13.324 ***
Control group	52	2.79 ± 0.605			2.54 ± 0.851			2.46 ± 0.851					
Abstraction													
Intervention group	51	1.57 ± 0.700	−0.089	0.929	1.78 ± 0.461	−2.294	0.024 *	1.78 ± 0.461	−3.261	0.002 **	4.726 *	3.988 *	4.726 *
Control group	52	1.56 ± 0.539			1.56 ± 0.539			1.46 ± 0.541					
Delayed Recall													
Intervention group	51	1.41 ± 0.983	1.349	0.181	2.45 ± 1.301	−4.007	0.000 ***	2.18 ± 1.144	−3.141	0.002 **	16.319 ***	6.309 *	14.549 ***
Control group	52	1.63 ± 0.658			1.63 ± 0.658			1.60 ± 0.664					
Orientation													
Intervention group	51	5.47 ± 0.833	−1.489	0.139	5.59 ± 0.669	−2.289	0.024 *	5.67 ± 0.589	−2.606	0.011 *	2.221	4.895 *	0.862
Control group	52	5.19 ± 1.049			5.19 ± 1.049			5.25 ± 0.988					

* *p* < 0.05, ** *p* < 0.01, *** *p* < 0.001.

## Data Availability

The raw data supporting the conclusions of this article will be made available by the authors, without undue reservation.

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
