# Peer review of "Motor–Cognitive Interventions May Effectively Improve Cognitive Function in Older Adults with Mild Cognitive Impairment: A Randomized Controlled Trial"

_behavsci, 2023, doi:10.3390/bs13090737_

Round 1

Reviewer 1 Report

This study aimed to investigate the impact of a motor-cognitive intervention on the cognitive function of older adults with Mild Cognitive Impairment (MCI) and sought to design a culturally appropriate approach for Chinese individuals. The primary objective was to determine whether the intervention could lead to improvements in cognitive abilities compared to a control group. The study findings revealed that participants in the intervention group exhibited significant cognitive improvement at the conclusion of the intervention as well as during the follow-up period, when compared to the control group. The authors concluded that the exercise-cognitive intervention holds promise for enhancing cognition in older adults with MCI. The paper is well-written and well-structured; however, it is noted that some aspects of the study may not be entirely novel, as they draw upon earlier research.

In light of these findings, I have several comments and questions:

1, The study's Table 2 indicates that the experimental group received psychological health education, physical-motor activities, and integrated cognitive training, suggesting that they were exposed to a familiar environment, trainers, examiners, and frequent use of psychological terminology. Conversely, the control group was exposed to fewer factors mentioned above. These elements could potentially influence the MoCA scores of the participants. It would be beneficial for the authors to discuss how they addressed or controlled for these potential confounding factors to ensure the validity and reliability of their results.

2, The study mentions an IRB Number of 20230309, which seems to be a date. However, it is also mentioned that the project included a 3-month intervention and a 3-month follow-up. This raises questions about how the authors managed to complete the entire project within a 4-month timeframe.

3, The authors did not present the MMSE scores at T2 and T3, which could be relevant for better understanding the cognitive changes over time.

4, The follow-up period did not include an assessment of whether the participants progressed to Alzheimer's disease.

5, To improve the presentation of the data, it is suggested to include error bars in the figures.

Addressing these comments and questions will contribute to the overall strength and comprehensiveness of the study and help readers better interpret and evaluate the findings.

Reviewer 2 Report

The aim of this study is very interesting, because focus on a motor-cognitive intervention to improve cognitive function in older adults with mild cognitive impairment, which is especially important due to the progressive growth of the elderly population.

However, some suggestions are indicated below in order to improve some sections of the manuscript:

- Cognitive function is assessed by MoCa, that is a screening test to detect MCI. It would have been interesting to include a more complete neuropsychological assessment of different cognitive domains to determine more precisely the benefits of the intervention.

- The size of the sample is small and may be not representative of the population, although it should be considered that it is an intervention study

- We suggest to include more recent studies from 2020, since the study on this topic has been recently investigated. We suggest to include this reference:

Comparative Cognitive Effects of Choreographed Exercise and Multimodal Physical Therapy in Older Adults with Amnestic Mild Cognitive Impairment: Randomized Clinical Trial

https://pubmed.ncbi.nlm.nih.gov/31868666/

- We also suggest synthesizing the information included in the article and reducing its length as much as possible.

Reviewer 3 Report

This is an interesting study on the effects of an intervention that utilizes various stimulation domains that include motor and cognitive training. The intervention (motor-cognitive) was performed in total 6 times (once every 2 weeks (fortnight) for 12 weeks) with baseline and 2 follow-up measurements of; MMSE, ADL, and MoCA. The authors detected differences in total MoCA scores, spatial-visual / executive, naming scores, language, extraction, and delayed recall, between controls and intervention groups, and over time.

I am unaware of any intervention performed for such a short period of time (6 sessions) that produces such a large change in MoCA scores and subcategories. The author needs to fully explain this and compare and contrast it with other interventions. This point must be included in the discussion.

Comments:

-          CONSORT study flow chart and table should be included.

Exercise Program

-          This reviewer had a very difficult time following the actual interventions(motor-cognitive) performed. This could be better explained using figures of the protocol and flow diagram

-          Please compare this intervention with any other CLINICAL intervention only performed 6 times or less in which there have been such significant differences

-          What are the METS (Metabolic Equivalents) or oxygen consumption produced by the MOTOR intervention?

Data Analysis

What is the actual effect size of the intervention on MMSE, and total MoCA. ?

Was there a power analysis done to determine the number of subjects? Please explain in detail

Consider putting Fig 1,2,3,4,5,6 into ONE Figure using Panels.  

Table 4 should probably best be displayed in landscape

Minor spelling errors and context

Round 2

Reviewer 1 Report

Comments have been addressed.

Author Response

Dear Reviewer,

First, thank you very much for taking the time to carefully review my paper in your busy work. Thank you for any comments and suggestions, these valuable comments and suggestions play a very important role in improving the content and structure of the article. I deeply admire your rigor, professionalism, patience, and detail. It is your review work that has made an important contribution to the improvement of the quality of my paper.

Thank you again for your support. Wish you all the best!

Sincerely,

Jinxuan Cheng